# Multi-Objective Optimizations of Non-Isothermal Simulated Moving Bed Reactor: Parametric Analyses

**Jian Wang [1], Wenwei Chen [1], Yan Li [2], Jin Xu [1,\*], Weifang Yu [1,\*] and Ajay K. Ray [2,\*]**

[1]  Chemical Engineering Department, Wenzhou University, University Town, Wenzhou 325035, Zhejiang, China; wangjian19931201@foxmail.com (J.W.); chenwenwei1717@163.com (W.C.)

[2]  Department of Chemical and Biochemical Engineering, Western University, London, ON N6A 5B9, Canada; yli2589@uwo.ca

\*  Correspondence: xujin@wzu.edu.cn (J.X.); ywf@wzu.edu.cn (W.Y.); aray@eng.uwo.ca (A.K.R.)

**Abstract:** Simulated moving bed reactor (SMBR), a multicolumn multifunctional integrated reactor system, which can be exploited with on-site adsorptive separation to enhance conversion of equilibrium-limited reversible chemical reaction. In this article, for generality, a dimensionless SMBR model was developed and effects of five representative temperature distributions among different zones on the performance of an SMBR for reversible reaction in the general form of a reactant decomposed to two products were evaluated based on simultaneous maximization of unit throughput and product purity. Multipliers were applied to adjust some of the model parameters such that different operation modes can be compared under various conditions in the parametric space. The multiobjective optimization problems were solved using the non-dominated sorting genetic algorithm. All simulations were carried out using FORTRAN codes. The results showed that both kinetics and adsorptive separation play important roles in SMBR. When kinetics is fast or adsorptive potency of the reactant is higher than the desired product (B) but lower than byproduct (C), non-isothermal operation can significantly improve unit throughput. On the contrary, feed concentration and reaction enthalpy have minor effects on the optimal solutions. Decision variables based on flow rate ratios and internal concentration profiles were used to explain the trends of Pareto optimal solution.

**Keywords:** simulated moving bed reactor; non-isothermal; multiobjective optimization; parametric sensitivity; Pareto



## 1. Introduction

Simulated moving bed (SMB) system consists of several series-connected packed columns that are typically divided into four zones by the two inlet ports, feed and desorbent, and two outlet ports, raffinate and extract [1–3]. Continuous countercurrent adsorptive separation can be mimicked by periodically switching the inlet and outlet ports along the mobile phase flow direction. The operation zones play different roles in a properly designed operation [1,3,4].

A SMB unit can be used to carry out chemical reactions, forming a SMB reactor (SMBR) [5,6]. Its applications on the intensification of various reactions, such as esterification [7–10], acetalization [11], etherification [12], hydrogenation [13,14], isomerization [15,16], production of sugar [17,18] and p-xylene [19,20] have been reported. Efficient in-situ separation of reactants and/or products is crucial for the conversion enhancement of a SMBR beyond thermodynamic equilibrium. It has been shown that several recent modifications of SMB, such as VariCol, PowerFeed and ModiCon, that further enhance in-situ separation for difficult separation such as chiral drugs can also be effectively applied to SMBR [21–24].

Conventional SMB for separation and SMBR for reactive processes are isothermally and isocratically operated. Several studies showed that temperature or concentration

gradients might be applied to increase the unit throughput of SMB separation system by adjusting the adsorption strength of each zone according to its functional roles [25–29]. The ongoing study carried out in this research group has been aimed to comprehensively evaluate the introduction of non-isothermal operation in a SMBR. It is to be noted that in our work the non-isothermal operation is applied on purpose, and is therefore, different from the thermal effects induced by inherent reaction enthalpy and limited heat transfer rate [30].

Effects of non-isothermal operation on a 4-zone SMBR for the synthesis of methyl acetate catalyzed by Amberlyst15 have been investigated under optimal conditions in our previous studies [31,32]. In the first non-isothermal article [31], extensively used "Equilibrium Theory" [33,34] for the design of SMB separation processes followed by "restrictive optimization" [35] was applied to maximize the unit throughput for practically complete conversion and complete separation. Moreover, maximum flowrate was assigned to Zone I and flowrate ratios (typically expressed as $m$-values) in Zones I and IV were conservatively fixed (see Figure 1). The search for suitable operating conditions were restricted to the $(m_{II}–m_{III})$ plane. In the subsequent non-isothermal article [32], "non-restrictive" multiobjective optimization [35] was carried out allowing reduced purity requirement, solvent consumption, and product yield into consideration. Optimal operating conditions for various optimization problems were identified in the parametric space consisting of all $m$-values. Since methanol, one of the reactants, is used as the mobile phase and in large excess, methyl acetate synthesis in our studies [31,32] falls in the catalogue of $A \leftrightarrow B + C$ reactions. This specific model reaction has the following features: (i) the reaction/separation system is kinetically controlled; (ii) the desired product (methyl acetate) is less adsorbed than the byproduct (water) and is collected at the raffinate port; (iii) reactant (acetic acid) has an adsorption strength close to the less adsorbed product (the ester) and (iv) reaction rates are more sensitive to temperature than adsorption equilibrium constant.

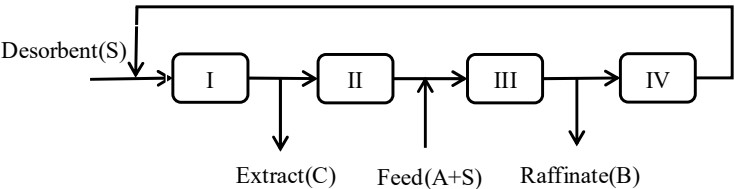

**Figure 1.** Schematic diagram of simulated moving bed reactor (SMBR) for a $A \leftrightarrow B + C$ reaction.

In this article, the scope of the work is not limited to a specific chemical reaction but is extended to any reversible reactions in the form of $A \leftrightarrow B + C$. The objective of this study is to evaluate the feasibility of application of temperature gradient in the SMBR system. More specifically, it is aimed at finding answers to the following two questions, which are of great academic and industrial interests: (a) What kind of kinetic and equilibrium properties should a reversible reaction of type $A \leftrightarrow B + C$ must have such that the non-isothermal operation mode can enhance the performance of a SMBR, and (b) how to adjust the operating parameters to meet the required objectives (productivity, purity, etc.) during the design of a non-isothermal SMBR process? To achieve these objectives, for the first time, a dimensionless mathematical model of SMBR for reversible reactions in the form of $A \leftrightarrow B + C$ was developed for the simulation of non-isothermal SMBR processes. Subsequently, for more generality, effects of reaction rate, adsorption strength, activation energy, feed concentration and reaction enthalpy on the performance among various SMBR operation modes with different temperature distributions were systematically investigated based on multiobjective optimization results. To the best of our knowledge, this is the first attempt to apply the dimensionless model and parametric analysis on the multiobjective optimization of non-isothermal SMBR processes. Hence, the results presented in this article is not restricted to methyl acetate synthesis but is valid for any reactions in the catalogue of $A \leftrightarrow B + C$. Furthermore, to provide deep insights to the trends of obtained optimization results and corresponding operating variables,

additional informative results, including internal concentration profiles, conversion and reaction rates in different zones, are discussed in detail.

## 2. Modeling of SMBR

Mathematical model to describe non-isothermal 4-zone SMBR and numerical schemes used in this study were developed and described in detail in our previous study [31,32]. Some of the model parameters used in the model were experimentally measured by Yu et al. [36]. In this article, the model is converted into dimensionless forms as described below.

### 2.1. Mathematical Model

The reaction considered in this work can be generalized as

$$A \leftrightarrow B + C \tag{1}$$

where *A*, *B* and *C* are the reactant, desired product and byproduct, respectively. Figure 1 shows the schematic diagram of a non-isothermal 4-zone SMBR for the reversible reaction $A \leftrightarrow B + C$. In our previous publications [31,32], the conventional equilibrium-dispersion (ED) model had been extended to account for the catalytic reaction. Due to the simplification of negligible radial gradients and instantaneous adsorption equilibrium, the reactor model is 1-dimensional. In addition, the linear isotherm for all species was assumed. For generality, the following dimensionless model was developed in this work using dimensionless variables listed in Table 1. For easy references, both dimensional and dimensionless variables are summarized in the nomenclature.

**Table 1.** Dimensionless model variables.

| Variable | Definition | Variable | Definition |
|:---:|:---:|:---:|:---:|
| $\varphi$ | $\frac{1-\varepsilon}{\varepsilon}$ | $x$ | $\frac{c}{c^0}$ |
| $\tau$ | $\frac{4tQ^0_{max}}{\pi d^2 L\varepsilon}$ | $\widehat{Q}$ | $\frac{Q}{Q^0_{max}}$ |
| $Z$ | $\frac{z}{L}$ | $Pe$ | $\frac{4LQ^0_{max}}{\pi d^2 \varepsilon D_{app}}$ |
| $\widehat{r}$ | $\frac{r\pi d^2 L\varepsilon}{4Q^0_{max}c^0}$ | $\widehat{\lambda}$ | $\frac{\lambda\pi d^2 L\varepsilon}{4Q^0_{max}}$ |
| $\theta$ | $\frac{T-T_{min}}{T_{max}-T_{min}}$ | $\theta^{rel}$ | $\frac{T_{min}}{T_{max}-T_{min}}$ |
| $Da$ | $\frac{k_f\pi d^2 L\varepsilon}{4Q^0_{max}}$ | $\widehat{K}_{eq}$ | $\frac{K_{eq}}{c^0}$ |
| $e_f$ | $\frac{E_f}{R(T_{max}-T_{min})}$ | $\Delta h_{rxn}$ | $\frac{\Delta H_{rxn}}{R(T_{max}-T_{min})}$ |
| $\Delta h_{ads}$ | $\frac{\Delta H_{ads}}{R(T_{max}-T_{min})}$ | | |

Component mass balance:

$$(1+\varphi H_i)\frac{\partial x_{i,j}}{\partial \tau} + \varphi x_{i,j}\frac{\partial H_i}{\partial \theta_j}\frac{\partial \theta_j}{\partial \tau} + \widehat{Q}_j\frac{\partial x_{i,j}}{\partial Z} - \frac{1}{Pe_{i,j}}\frac{\partial^2 x_{i,j}}{\partial Z^2} - \varphi v_i \widehat{r} = 0 \tag{2}$$

where *x* and *θ* are dimensionless concentration and temperature, *φ* is phase ratio, *H* is Henry's constant, *τ* and *Z* are dimensionless time and axial coordinates, *θ* is temperature normalized between 0 and 1, $\widehat{Q}$ and $\widehat{r}$ are the flowrate and reaction rate, *Pe* is the Peclet number, *v* is the stoichiometric number and *i* (=*A*, *B*, *C*) and *j* (=*I*, *II*, *III*, *IV*) are indices of components and columns (zones), respectively. Effects of composition, conversion and temperature on the flowrate were neglected. $\widehat{Q}$ is therefore constant in each column.

Reaction rate was described based on solid phase volume.

$$\widehat{r} \;=\; Da\left( H_A x_A - \frac{H_B H_C x_B x_C}{\widehat{K}_{eq}} \right) \tag{3}$$

where $Da$ is Damkohler number and $\widehat{K}_{eq}$ is equilibrium constant. $H$, $Da$ and $\widehat{K}_{eq}$ were assumed to be temperature dependent. They are defined based on the reference temperature of $\theta^{ref} = 0.667$, consistent with the previous publication [32].

$$H_i \;=\; H_i^{ref} \exp\left[ \Delta h_{ads} \left( \frac{1}{\theta^{rel} + \theta^{ref}} - \frac{1}{\theta^{rel} + \theta} \right) \right] \tag{4}$$

$$Da \;=\; Da^{ref} \exp\left[ e_f \left( \frac{1}{\theta^{rel} + \theta^{ref}} - \frac{1}{\theta^{rel} + \theta} \right) \right] \tag{5}$$

$$\widehat{K}_{eq} \;=\; \widehat{K}_{eq}^{ref} \exp\left[ \Delta h_{rxn} \left( \frac{1}{\theta^{rel} + \theta^{ref}} - \frac{1}{\theta^{rel} + \theta} \right) \right] \tag{6}$$

where $\theta^{rel}$ is a constant defined in Table 1.

Exponential temperature transition of a column after a switch [25,31]

$$\theta_j \;=\; \theta_j^{\infty} + \left( \theta_{jpre}^{\infty} - \theta_j^{\infty} \right) \exp\left( -\widehat{\lambda} \tau \right) \tag{7}$$

where $\theta_j^{\infty}$ is the final temperature of the current switch preset for column $j$, $\tau$ is reset to 0 after each switch operation,

$$jpre \;=\; \begin{cases} I & j = IV \\ j+1 & j = I, II, III \end{cases} \tag{8}$$

Criterion for increasing adsorption strength gradient from Zone I to Zone IV was imposed.

$$\theta_I^{\infty} \geq \theta_{II}^{\infty} \geq \theta_{III}^{\infty} \geq \theta_{IV}^{\infty} \tag{9}$$

The above equations are consistent with previous publications [31,32]. It is acknowledged that the model had been developed based on several assumptions, especially the simple description of column temperature. However, these simplifications should not have qualitative effects on the conclusions in this study. More detailed description of the model can be found elsewhere [31].

### 2.2. Numerical Solution

Partial differential Equation (2) was discretized along the axial direction using the Martin Synge method [37]. Martin Synge method divides a column into N equally spaced sections and uses the 1st-order backward approximation for convection term $(\partial/\partial z^2)$. If $N$ is properly chosen, the truncation error can be used to eliminate the dispersion term $(\partial/\partial z^2)$. The dispersion term $(\partial/\partial z^2)$ is replaced by the truncation error introduced by the 1st order backward approximation of convection term $(\partial/\partial z)$,

$$\frac{\Delta Z}{2} \left.\frac{\partial^2 x_{i,j}}{\partial Z^2}\right|_M \;=\; \left.\frac{\partial x_{i,j}}{\partial Z}\right|_M - \frac{x_{i,j,M} - x_{i,j,M-1}}{\Delta Z} + o\left( \Delta Z^2 \right) \tag{10}$$

$$\left.\frac{\partial x}{\partial Z}\right|_j \;=\; \frac{x_{j+1} - x_{j-1}}{2\Delta Z} \Delta Z \;=\; \frac{2}{Pe} \;=\; \frac{1}{N} \tag{11}$$

where $\Delta Z$ is the equally spaced step size, $M$ is the index of mesh points. Plate number $N$ was assumed constant for all components in all columns. Substituting Equation (10) to Equation (2) gives

$$(1 + \varphi H_i)\frac{\partial x_{i,j,M}}{\partial \tau} = -\varphi x_{i,j,M}\frac{\partial H_i}{\partial \theta_j}\frac{\partial \theta_j}{\partial \tau} - \widehat{Q_j}\frac{x_{i,j,M} - x_{i,j,M-1}}{\Delta Z} + \varphi v_i \widehat{r} \qquad (12)$$

Due to the elimination of 2nd order derivatives, only the inlet condition is retained for each column and is given by the following node balance.

$$\text{Desorbent Node}: \ x_{i,I,M=0} = \frac{\widehat{Q_{IV}}x_{i,IV,M=N}}{\widehat{Q_I}} \qquad (13)$$

$$\text{Extract Node}: \ x_{i,II,M=0} = x_{i,I,M=N} \qquad (14)$$

$$\text{Feed Node}: \ c_{i,III,M=0} = \frac{\widehat{Q_{II}}x_{i,II,M=N} + \left(\widehat{Q_{III}} - \widehat{Q_{II}}\right)x_{i,feed}}{\widehat{Q_{III}}} \qquad (15)$$

$$\text{Raffinate Node}: \ x_{i,IV,M=0} = x_{i,III,M=N} \qquad (16)$$

Equation (7) can be analytically solved and directly substituted into Equation (12), which is then simplified to 1st-order ordinary differential equations for $x$ values. Rigorously, cyclic condition applied as

$$x_{i,j,M}(\tau) = x_{i,j,M}(\tau + \tau_s) \qquad (17)$$

where $\tau_s$ is the switching time and $j$ and $M$ are from I to IV and 1 to $N$, respectively. To avoid the difficulty in numerically solving the equations with cyclic conditions, the following initial conditions were used.

$$x_{i,j,M}(\tau = 0) = 0 \qquad (18)$$

Equation (12) supplemented with Equation (18) is in the form of initial value problems (IVPs). LSODE package [38] was used in this work for the integration. A minimum number of 15 cycles (60 switches) and relative mass balance error of less than $5 \times 10^{-3}$ were used to justify the cyclic steady-state.

$$ERR = \frac{\int_0^{\tau s}\left[\left(\widehat{Q_{III}} - \widehat{Q_{IV}}\right)\sum_i w_i x_{i,III,N} + \left(\widehat{Q_I} - \widehat{Q_{II}}\right)\sum_i w_i x_{i,I,N}\right]d\tau}{\left(\widehat{Q_{III}} - \widehat{Q_{II}}\right)x_{A,feed}\tau_s} \leq 5 \times 10^{-3} \quad (19)$$

where $w$ equals to 1 for $A$ and 0.5 for $B$ and $C$.

In all, the original partial differential equation was first discretized by the Martin–Synge method. The rigorous cyclic conditions were replaced with the initial conditions. As such, the model was converted to IVP and then solved using well established LSODA software. More details can be found elsewhere [31,32].

### 2.3. Model Parameters

Dimensionless model parameters are summarized in Table 2. These values are consistent with the dimensional ones used in previous publications [31,32,36], which are separately provided in Supplementary Materials as Table S1. In this work, parametric sensitivity analyses were carried out to investigate the effects of several parameters on optimal performance of various modes of SMBR operations. Parametric studies have been widely used in the design and analyses of chemical reactors [39,40]. Following the strategy suggested by Xu et al. [41], multipliers were applied to adjust some of the parameters. Provided in Table 2 are the original values of these parameters, corresponding to the multipliers equal to 1. In a previous work [29], sensitivity of $m_{IV}$, an operating variable, which

has no obvious trends when unit throughput and product purity were simultaneously maximized was applied to reduce solvent consumption. The sensitivity study in this work, however, was carried out for some of the kinetic and equilibrium properties.

**Table 2.** Dimensionless model parameters.

| Column and Operation | | | Adsorption Equilibrium | | | Kinetics | |
| --- | --- | --- | --- | --- | --- | --- | --- |
| $\varphi$ | 1.5 | | $H^{ref}$ | $\Delta h_{ads}$ | $Da_f^{ref}$ | 2.48 ($\alpha_1$) * | |
| $N$ | 50 | $A$ | 0.426 ($\alpha_2$) | −157.5 | $e_f$ | 354.4 ($\alpha_3$) | |
| $x_{A,feed}$ | 1.0 ($\alpha_4$) | $B$ | 0.375 | −73.0 | $K_{eq}^{ref}$ | 167.4 | |
| $\theta$ | $0 \leq \theta^\infty \leq 1$ | $C$ | 2.92 | −68.4 | $\Delta h_{rxn}$ | −46.8 ($\alpha_5$) | |

*: Parameters subjected to modifications with corresponding multipliers in brackets. For example, $Da_f^{ref} = 2.48 \times \alpha_1$

## 3. Multiobjective Optimization of an SMBR

### 3.1. Variables

A 4-zone SMBR has five independent operating parameters, i.e., switching time and flowrates in the four zones, similar to conventional SMB for separation. Maximum flowrate is assigned to Zone I, which imposes the maximum pressure drop within the system. Temperature dependence of viscosity and maximum flowrate was considered by the following correlation [32,42].

$$\widehat{Q}_I = \exp\left[86.7 \times \left(\frac{1}{\theta^{rel} + \theta^{ref}} - \frac{1}{\theta^{rel} + \theta_{II}^\infty}\right)\right] \tag{20}$$

Since $Q_I$ can be determined by the preset temperature distribution, the number of independent operating variables subjected to optimization is reduced to 4. They were described by the following flowrate ratio.

$$m_j = \frac{\widehat{Q}_j \tau_s - 1}{\varphi} \tag{21}$$

In the case of non-isothermal operations, temperature distribution among the four zones is also adjustable and is therefore equivalent to another variable. For simplicity, it was assumed that the preset temperature of each zone is limited to four values equally spaced from 0 to 1. Five representative cases satisfying the criterion in Equation (9) were considered. As compiled in Table 3, Cases 1 and 5 are isothermal operations at the highest and lowest temperature, respectively; Case 2 has a highest temperature difference between Zones III and IV; Case 3 has a highest temperature difference between Zones II and III, which, according to "Equilibrium Theory" [33], is favorable to unit throughput for a separation process; Case 4 has a unique distribution with temperature monotonously descending from I to IV. Theoretically, the operation mode can be treated as a discrete variable with a limited number of values and simultaneously subjected to optimization with other operating parameters [43]. However, to simplify the numerical calculations and get more comprehensive results, operations with various temperature distributions were individually optimized and then compared in this work.

**Table 3.** The five representative cases.

| No. | $\theta^{\infty}$ * | | | |
| --- | --- | --- | --- | --- |
| | **I** | **II** | **III** | **IV** |
| 1 | 1 | 1 | 1 | 1 |
| 2 | 1 | 1 | 1 | 0 |
| 3 | 1 | 1 | 0 | 0 |
| 4 | 1 | 0.667 | 0.333 | 0 |
| 5 | 0 | 0 | 0 | 0 |

*: $\theta^{\infty}$ is the preset temperature in Equation (7).

### 3.2. SMBR Performance

Product purity (*PurB*), yield (*YB*) and unit throughput (*UT*) were used to evaluate SMBR performance. They were defined as below.

$$PurB = \frac{\int_0^{\tau_s} x_{B,III,N}d\tau}{\int_0^{\tau_s} \sum_i x_{i,III,N}d\tau} \tag{22}$$

$$YB = \frac{\left(\widehat{Q}_{III} - \widehat{Q}_I\right)\int_0^{\tau_s} x_{B,III,N}d\tau}{\left(\widehat{Q}_{III} - \widehat{Q}_{II}\right)x_{A,feed}\tau_s} \tag{23}$$

$$UT = \widehat{Q}_{III} - \widehat{Q}_{II} \tag{24}$$

In addition, conversion in each individual zone defined below will be involved in the discussion section.

$$Conv_j = \frac{\int_0^{\tau_s}\int_0^1 \hat{r}_j dZd\tau}{\hat{Q}_{feed}x_{A,feed}\tau_S} \tag{25}$$

### 3.3. Definitions of Optimization Problems

In general, high unit throughput, reduced solvent consumption, desired purity and recovery (yield) are the major objectives during the design of an SMBR process. One can consider many different configurations of SMBR for the multiobjective optimization study. Several optimization problems with different combination of selections of objectives were investigated in our previous work [32]. This article is mainly focused on simultaneous maximization of product purity and unit throughput. In addition to these two objectives, constraints on purity and yield, greater than 0.95 and 0.90, respectively, were applied to practically narrow down the search space range in the operational range of parameters. A 4-zone SMBR has 5 independent operational parameters. In this work, flowrate in Zone I was fixed at the maximal flowrate, which is a scaling factor normally limited by column pressure [1,4]. Following the notions of "Equilibrium theory" that has been extensively applied in SMB design and analyses, the remaining 4 parameters were described by *m*-values (flow rate ratios) in the 4 zones. These *m*-values were all set to be the decision variables that can be independently tuned by an operator for simultaneous optimization of the selected objective functions. For clarity, the optimization problem is summarized in Table 4.

**Table 4.** Optimization problem.

| Objectives | Constraints | Variables | |
| --- | --- | --- | --- |
| | | **Decision** | **Fixed** |
| Max *PurB* | *PurE* > 0.95 | $m_I, m_{II}, m_{III}, m_{IV}$ | $\widehat{Q}_I$ (Equation (20)) |
| Max *UT* | *YE* > 0.90 | | |

Since the operating parameters have contradicting effects on these objectives [28], a set of solution points, called Pareto solutions [44,45], are normally obtained. Non-dominated sorting genetic algorithm (NSGA-II) [46] was applied to obtain the Pareto solutions. The genetic algorithm was shown as a versatile, flexible and robust optimization tool [47]. Population and generation numbers were both set to be 100. For easy reference, upper and lower bounds of variables, the key parameters required using NSGA, are provided in Supplementary Materials as Table S2. All calculations were programmed in FORTRAN codes and performed on Lenovo ThinkPad L440 personal computers equipped with 2.30 GHz Intel core i7 processors. Upon the presentation of Pareto solutions, some of the obviously off-trend points obtained by NSGA were manually removed.

## 4. Results and Discussion

Effects of five parameters, namely, $Da^0$, $H_A^0$, $e_f$, $x_{A,feed}$ and $h_{rxn}$ on the comparison among SMBR operations with different temperature distributions were investigated. Multiobjective optimizations were carried with one of the parameters modified by the corresponding multiplier. The other parameters were fixed at their original values in Table 2 (except the modification of $H_A^0$, see Section 4.2). The multipliers are compiled in Table 2. The results are presented and discussed below. The results with all multipliers equal to 1 have been reported in a previous publication [32]. For clarity and easy comparison, these results are also included.

### 4.1. Effects of the Forward Reaction Rate

A multiplier of $\alpha_1$ was used to modify $Da^0$. For each $\alpha_1$ value and each case with different temperature distribution, the optimization problem defined in Table 2 was individually solved by searching parametric space consisting of $m_I$, $m_{II}$, $m_{III}$ and $m_{IV}$.

#### 4.1.1. Pareto Solutions

Figure 2a shows the Pareto solutions for $\alpha_1 = 1$. For all cases, maximum unit throughput decreased with increased purity requirement, forming Pareto curves. It was shown that this model system was kinetically controlled in Zone III where a major fraction of the fed was converted [32]. As a result, the best Pareto curves were obtained for Cases 1 and 2, which had the same $\theta_{III}^\infty$ preset to be 1, highest temperature considered in our work.

Figure 2b–e shows the Pareto curves obtained for various $\alpha_1$ values. It is seen that, with increased $\alpha_1$, Pareto curves of all cases are shifted towards the right-hand side, i.e., towards higher unit throughput at given purity requirement increases. However, the significance of $\alpha_1$ effects on different cases is quantitatively different. Cases 1 and 2 exhibit the similar performance for all $\alpha_1$ values. At $\alpha_1$ between 3 and 4, their performance is surpassed by Case 3 with the highest temperature difference between Zones II and III. As $\alpha_1$ is increased to 10, Case 4 is comparable with Cases 1 and 2. The superiority of Cases 3 and 4 with different temperatures in Zones II and III becomes more significant with $\alpha_1$ further increased to 100. Case 5 isothermally operated at the lowest temperature always has the lowest unit throughput. As $\alpha_1$ equals 100, the Pareto curves approach those for pure separation (solid lines in Figure 1f), which were obtained by optimizing an SMB process for the separation of an equimolar $B/C$ mixture. The above trends of Pareto solutions at various $\alpha_1$ values suggest the combined effects of reaction kinetics and adsorptive separation. The system is kinetically controlled in the low $\alpha_1$ range and becomes dominated by separation as $\alpha_1$ increased to 100.

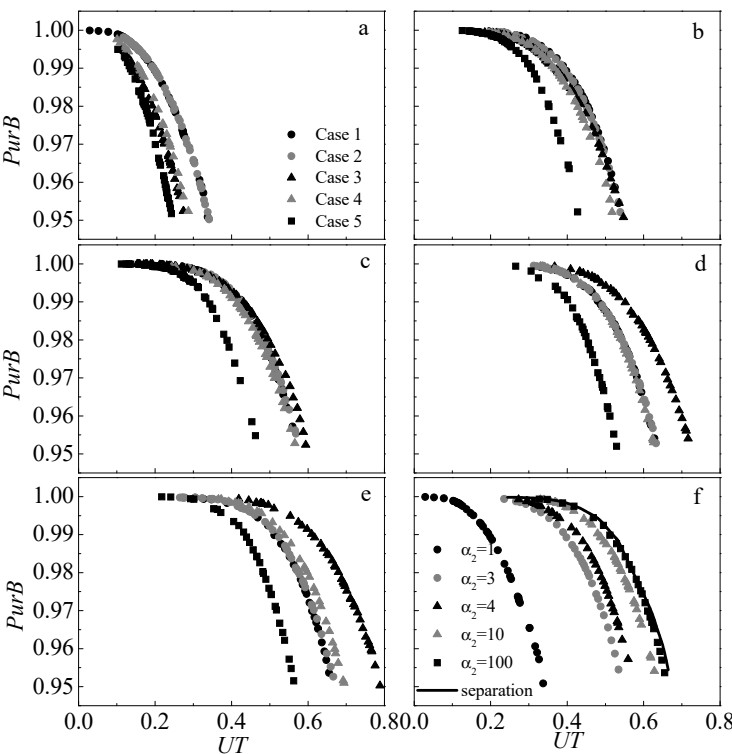

**Figure 2.** Effects of the forward reaction rate on Pareto solutions to simultaneously maximized *PurB* and *UT*. (**a**–**e**) are for $\alpha_1$ equal to 1, 3, 4, 10, 100, respectively; (**f**) is for Case 1 with various $\alpha_1$. The solid curve in f was for simulated moving bed (SMB) separation of an equimolar *B/C* mixture. Temperature distributions of the five representative cases are provided in Table 3.

4.1.2. Decision Variables: m Values

It may be straightforwardly derived from Equations (20)–(23) that

$$UT = \frac{m_{III} - m_{II}}{m_I + \varphi^{-1}} \widehat{Q}_I(\theta_{II}^\infty) \tag{26}$$

$m_I$, $m_{II}$ and $m_{III}$ for Pareto solutions of three cases (1, 3 and 5) with $\alpha_1$ values equal to 1 and 100 are shown in Figure 3. The upper channel is for $\alpha_1 = 1$ corresponding to a slow reaction rate. It may be seen that Cases 1 and 3 had similar trends of $m_I$, $m_{II}$ and $m_{III}$. More specifically, $m_I$ is maintained at the same level of about 0.35 for both cases at a low purity range and increases with further increased purity; $m_{II}$ is almost constant; $m_{III}$ decreases with increased purity in the lower range and becomes constant at higher purities, opposite to the trend of $m_I$. There are two quantitative differences in terms of m-values between Cases 1 and 3: (a) compared with Case 3, Case 1 had relatively lower $m_{II}$ and higher $m_{III}$, resulting in higher *UT* in the low purity range; and (b) mI of Cases 1 and 3 started to increase at PurB values of about 0.99 and 0.98 respectively. That $m_I$ of Case 1 started to increase at a higher purity contributes to its high *UT* at high purity range.

The lower channel of Figure 3 is for $\alpha_1$ equal to 100. The trends of $m$ values were similar to those of optimized conditions for separation SMB with less adsorbed species as the desired product [29,48]. Case 3, compared with Case 1, had higher *UT* for a given purity requirement. This is mainly attributed to its lower $\theta_{III}^\infty$ corresponding to increased adsorption strength, allowing for higher $m_{III}$ to sufficiently retain component *C*, the heavy byproduct. Both Cases 1 and 3 had the same $\theta_{II}^\infty$ preset at 1, the highest temperature. However, due to the model assumption of Equation (7), the column in Zone II experiences a temperature transition period after being switched from Zone III (see Figure S1 in Supplementary Materials). During this period, $\theta_{II}$ of Case 3 is lower than that of Case 1. Correspondingly, Case 3 had an overall higher adsorption strength in Zone II, allowing for

a lower $m_{II}$. However, the difference in $m_{II}$ between Cases 1 and 3 was much less significant than that in $m_{III}$. According to Equation (26), Case 1 had higher *UT* than Case 3.

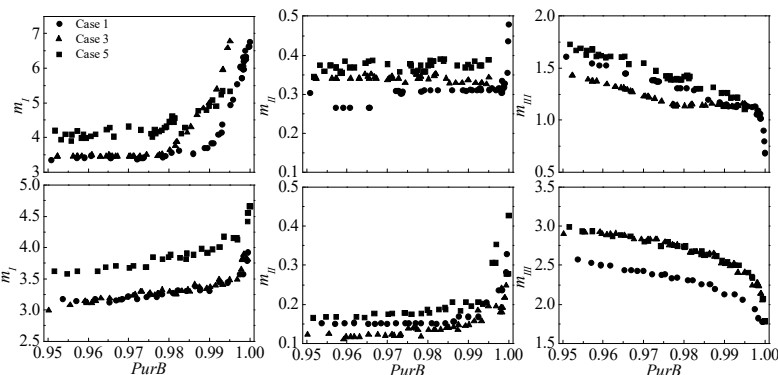

**Figure 3.** *m* values corresponding to simultaneously maximized *PurB* and *UT*. Cases 1, 3 and 5. Upper and lower channels are for $\alpha_1$ equal to 1 and 100; left, middle and right are for $m_I$, $m_{II}$ and $m_{III}$, respectively.

Compared with Case 1, Case 5 optimized for $\alpha_1$ = 100 had much higher $m_{III}$ and slightly higher $m_{II}$. Under the frame of "restrictive optimization" focused on the ($m_{II}$–$m_{III}$) plane [28], Case 5 should have higher unit throughput. However, due to the high $m_I$ and low $\widehat{Q_I}$, resulting from strong adsorption and high viscosity at low temperature, Case 5 isothermally operated at the lowest temperature still exhibited the lowest *UT*. These results verify the necessity of "non-restrictive optimization" [28] accounting for effects of switching time and maximum flowrate, even when only purity and unit throughput are taken into consideration.

For all cases, the corresponding $m_{IV}$ values are scattered in a relatively large window with no obvious qualitative trends. This agrees with Equation (26). Trends of optimized $m_{IV}$ are therefore omitted in the main manuscript but presented as Figure S2 in Supplementary Materials for completeness.

### 4.1.3. Internal Concentration Profiles and Reaction Rate

In order to further interpret the effect of reaction rate on the optimal unit throughput, internal concentration profiles of reactant *A* and byproduct *C* corresponding to *PurB* = 0.99, $\alpha_1$ = 1, Cases 1 and 3 are plotted in Figure 4. Profiles of product *B* that were not directly involved in the following discussion are provided in Supplementary Materials as Figure S3.

Figure 4a–c shows that the two cases had similar profiles of reactant *A*. In Zone III, *A* propagates toward raffinate port in the first 0.4–0.7$\tau_s$, and then its profile becomes essentially steady during the last fraction of a switch. It may be seen from Figure 5 that, as $\alpha_1$ equals 1, the reverse rate was significantly lower than the forward rate by about 2 magnitude orders, indicating successful separation of product *B* and byproduct *C* under optimized conditions. Due to the negligible reverse reaction, the forward (and overall) reaction rate exhibited the similar features as those of the profile of reactant *A*. Figure 5c,d also shows that, at $\alpha_1$ = 1, Case 3, compared with Case 1, has a lower reverse reaction rate. This may be attributed to the temperature difference between Zones II and III, which is favorable for the adsorptive separation of product *B* and byproduct *C* [49]. As shown in Figure 4b,d, byproduct C is about to breakthrough for both cases by the end of a switch cycle. However, as the system is kinetically controlled and the reverse reaction is negligible compared with the forward reaction, the efficient separation of Case 3 has only minor effects on the optimal unit throughput. As a result of the negligible reverse reaction rate, steady profile of *A* can be approached despite the development of the other components. Given the established steady profile of *A*, a further increase in $m_{III}$ is limited by the sufficient retaining of byproduct C that is more preferentially adsorbed than product B.

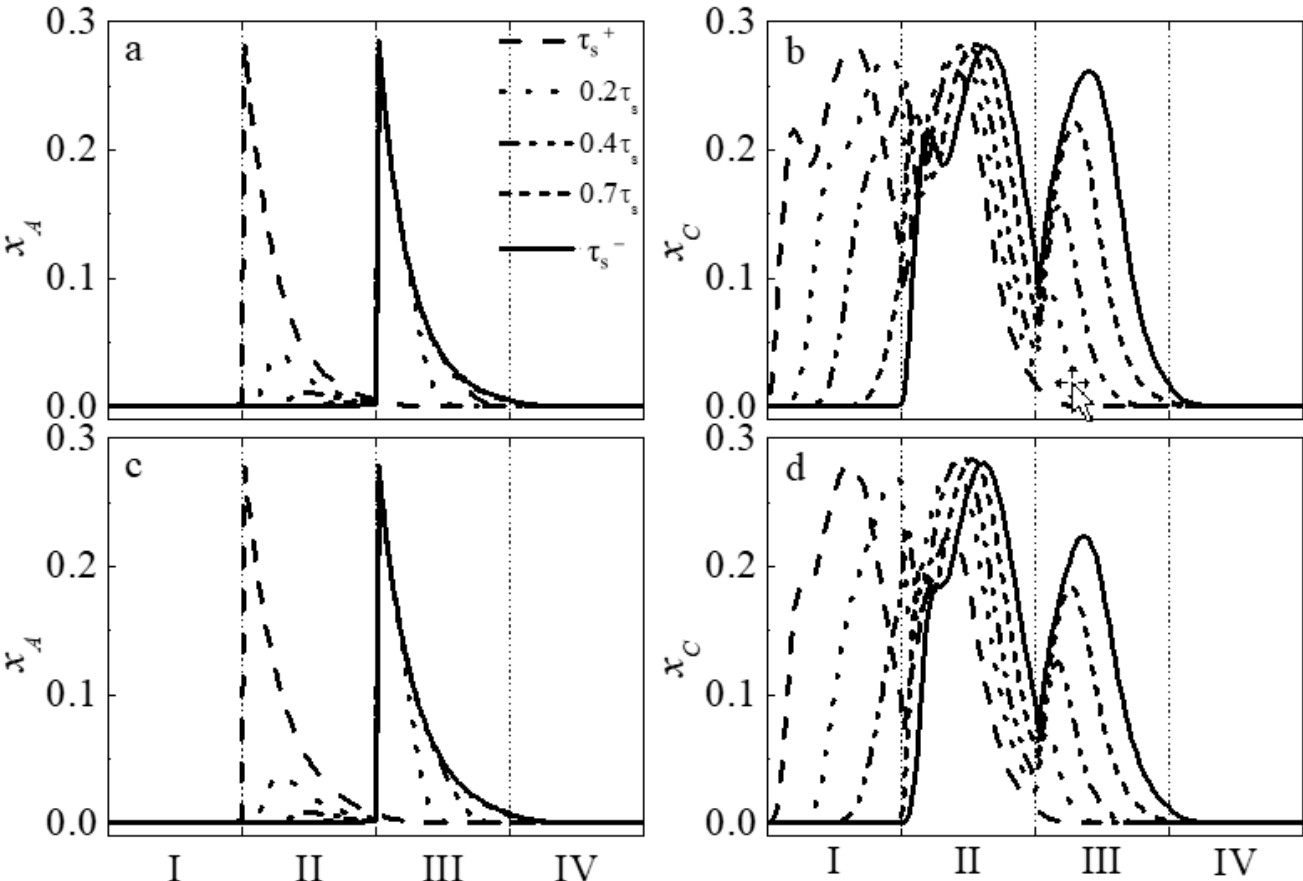

**Figure 4.** Internal concentration profiles of *A* and *C* corresponding to maximal *UT* at $\alpha_1 = 1$ and *PurB* = 0.99. (**a**): Cases 1, component *A*; (**b**): Cases 1, component *C*; (**c**): Cases 3, component *A*; (**d**): Cases 3, component *C*. $m_I$ to $m_{IV}$ values: (3.85, 0.31, 1.15, 0.31) for Case 1; (5.40, 0.33, 1.13, 0.36) for Case 3.

A noticeable difference between the two cases is observed in Zone I. As shown in Figure 4b–d, it takes a whole switch for Case 1 to purge out byproduct *C*, the preferentially adsorbed component, from Zone I, whereas the purge is essentially completed within about $0.7t_s$ for Case 3. In addition, as shown in Table 5, when all $\alpha$ values are set to 1, a major fraction about 85% of the conversion is realized in Zone III [34]. As shown in Figure 3, $m_I$ of Case 1 corresponding to 0.99 *PurB* is still at the platform of about 3.5 whereas that of Case 3 is increased to about 5.1. In order to explain the above comparison in $m_I$, Equation (12) is simplified by neglecting the temperature transition term.

$$(1 + \varphi H_i)\frac{\partial x_{i,j,M}}{\partial \tau} \approx -\widehat{Q_j}\frac{x_{i,j,M} - x_{i,j,M-1}}{\Delta Z} + \varphi v_i \widehat{r} \tag{27}$$

**Table 5.** Conversions in different zones.

| Case | $Conv_{II}$ $(10^{-2})$ | | $Conv_{III}$ $(10^{-2})$ | |
| :---: | :---: | :---: | :---: | :---: |
| | $\alpha_1 = 1$ | $\alpha_1 = 100$ | $\alpha_1 = 1$ | $\alpha_1 = 100$ |
| 1 | 13.0 | 0.5 | 86.2 | 99.2 |
| 2 | 13.4 | 0.5 | 85.9 | 99.3 |
| 3 | 17.2 | 0.5 | 81.8 | 99.2 |
| 4 | 16.3 | 0.5 | 82.7 | 99.3 |
| 5 | 15.0 | 0.5 | 84.1 | 99.2 |

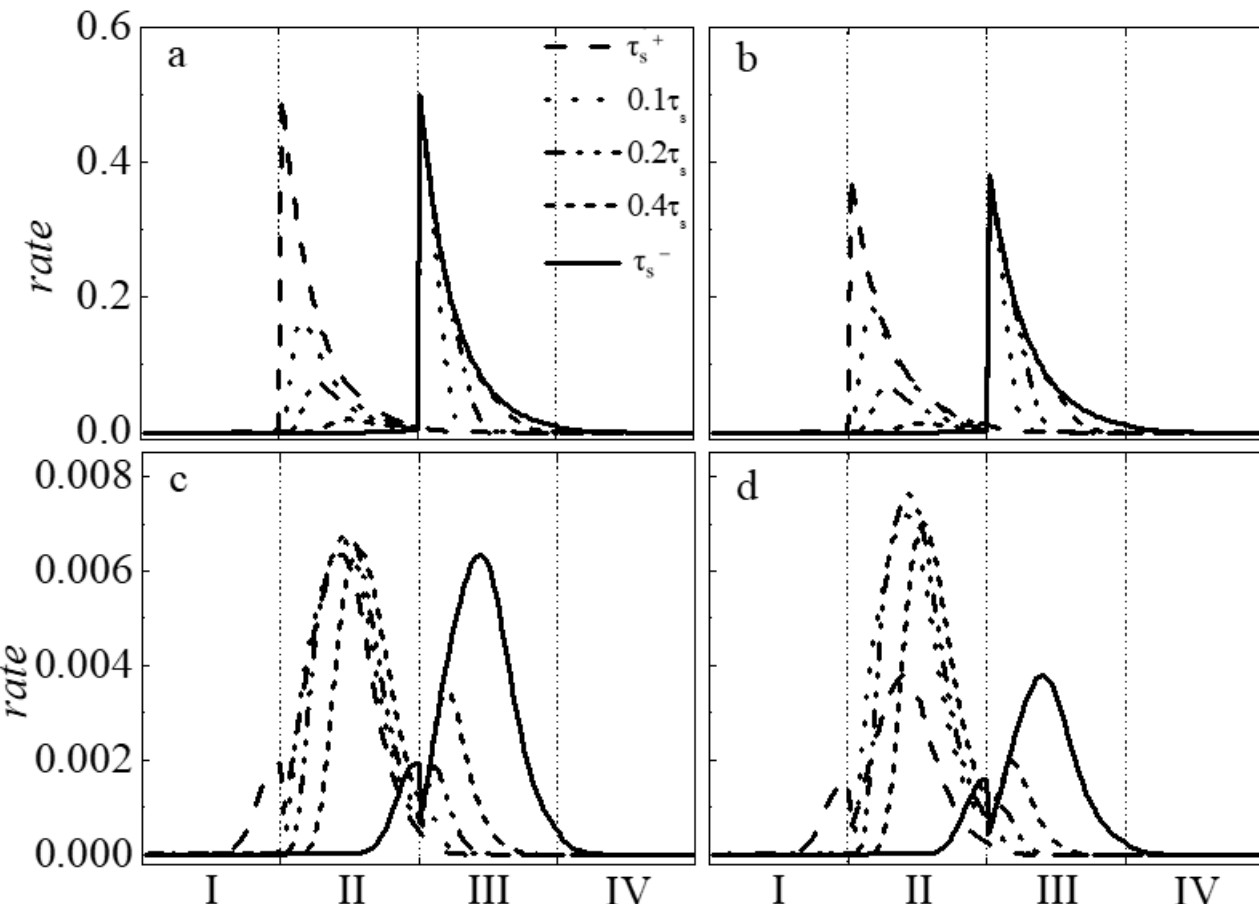

**Figure 5.** Forward and reverse reaction rates corresponding to maximal *UT* at $\alpha_1 = 1$ and *PurB* = 0.99. (**a**): Cases 1, forward; (**b**): Cases 3, forward; (**c**): Cases 1, reverse and (**d**): Cases 3, reverse.

Equation (27) is equivalent to a transient series-connected CSTR model. Applying definitions of *m* in Equation (21) and dimensionless time in Table 1 gives the estimation of residence time of reactant *A*.

$$\tau_j^{res} = \frac{\left[1 + \varphi H_A\left(\theta_j^{\infty}\right)\right] V_{col}/Q_j}{V_{col}\varepsilon/Q_{max}^0} = \frac{(1 + \varphi)\left[1 + \varphi H_A\left(\theta_j^{\infty}\right)\right](1 + \varphi m_I)}{\widehat{Q}_I\left(\theta_{II}^{\infty}\right)(1 + \varphi m_j)} \tag{28}$$

According to Equation (28) the residence time of *A* in Zone *j* increases with increased $m_j$. Compared with Case 1, Case 3 has a lower temperature and lower reaction rate in Zone III, where a major fraction of conversion is realized. Therefore, an increased $m_I$ (see Figure 3) is required to give a longer residence time in Zone III to sufficiently convert reactant *A*.

When $\alpha_1$ is increased to 100, more than 99% of the fed reactant is converted in Zone III (see Table 5). Figure 6 shows the concentration profiles and reaction rates of Cases 1 and 3 corresponding to $\alpha_1 = 100$ and *PurB* = 0.99. It is seen in Figure 6a,b that, under optimized conditions for both cases, reactant *A* was efficiently converted in Zone III due to the increased reaction rate. Concentration of *A* drops fast to the level corresponding to chemical equilibrium and 0 overall reaction rate (see Figure 6c,d). It is due to the separation of product *B* and byproduct *C* in Zone III that the concentration of *A* at equilibrium decreases while propagating towards the raffinate port. Comparison between Figure 6e,f shows that the reverse reaction rate of Case 3 was lower than that of Case 1 in Zone III, where the conversion was essentially completed, indicating that the former with a lower $\theta_{III}^{\infty}$ had more efficient on-site separation of *B* and *C*. At $\alpha_1$ increased to 100, the reaction

rate was fast and the system becomes separation-controlled. As a result, non-isothermal Case 3 with a temperature difference between Zones II and III had higher optimal *UT* than isothermal Case 1.

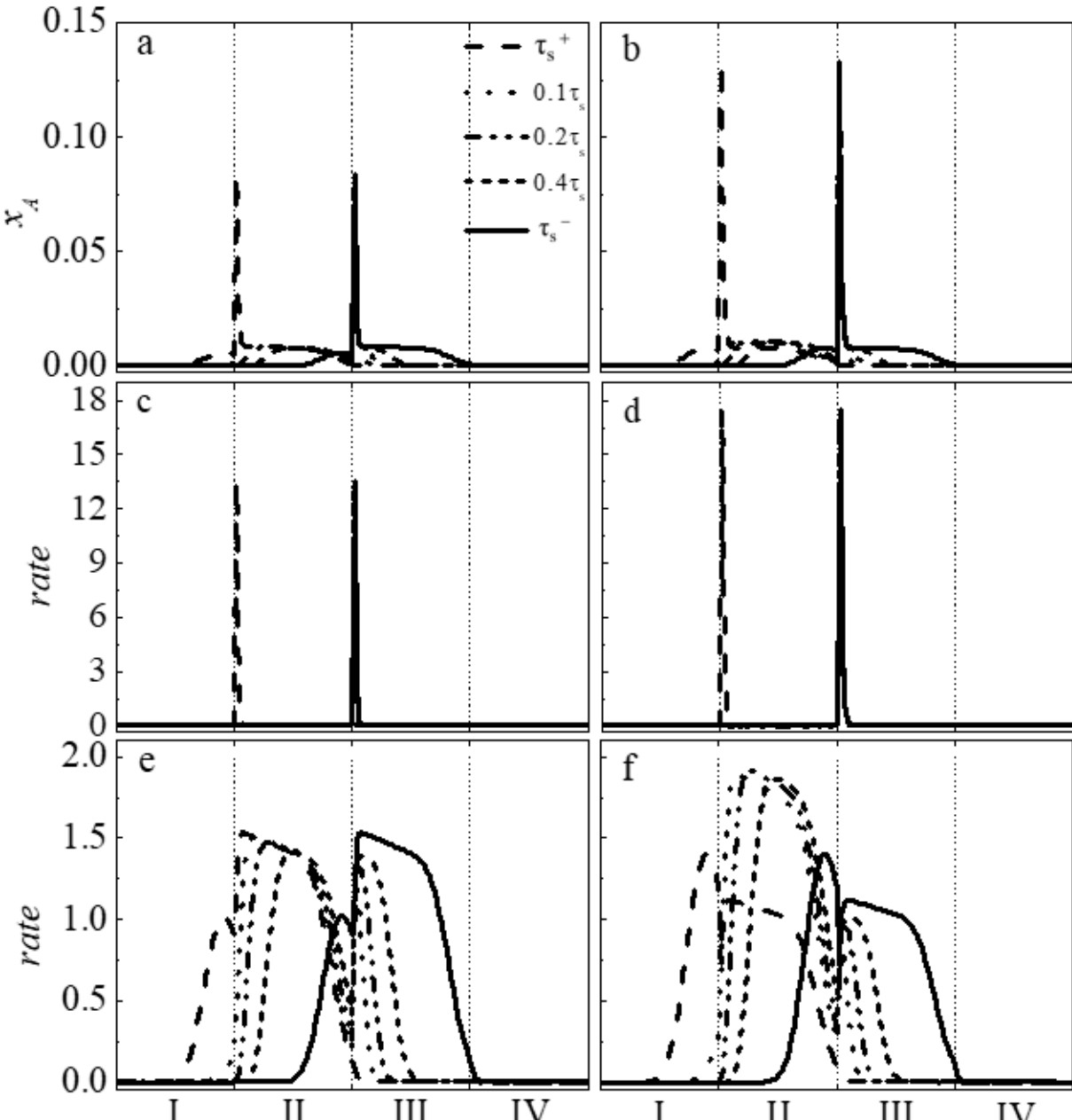

**Figure 6.** Concentration profiles and reaction rates corresponding to maximal *UT* at $\alpha_1 = 100$ and *PurB* = 0.99. (**a**): Cases 1, component A; (**b**): Cases 3, component A; (**c**): Cases 1, overall rate; (**d**): Cases 3, overall rate; (**e**): Cases 1, reverse rate; (**f**): Cases 3, reverse rate.

### 4.2. Effects of Adsorption Strength of Reactant A

Adsorptive separation plays an important role in a successfully designed SMBR. In the original model system of methyl acetate synthesis, reactant *A* has Henry's constants close to those of less adsorbed product *B*, i.e., $H_A \approx H_B < H_C$. In order to evaluate the effects of adsorption strength on optimal SMBR performance, a multiplier of $\alpha_2$ was used to adjust $H_A$. A total of 5 values, 0.1, 0.5, 4, 8 and 20, in addition to unity, were assigned to $\alpha_2$. As shown in the insert of Figure 7, the first two values correspond to the

condition of $H_A < H_B < H_C$, and the other three are for $H_B < H_A < H_C$, $H_B < H_A \approx H_C$, $H_B < H_C < H_A$, respectively. It is noted that, to retain the overall kinetic feature and thermodynamic equilibrium, both $Da$ and $\widehat{K}_{eq}$ were accordingly divided $\alpha_2$.

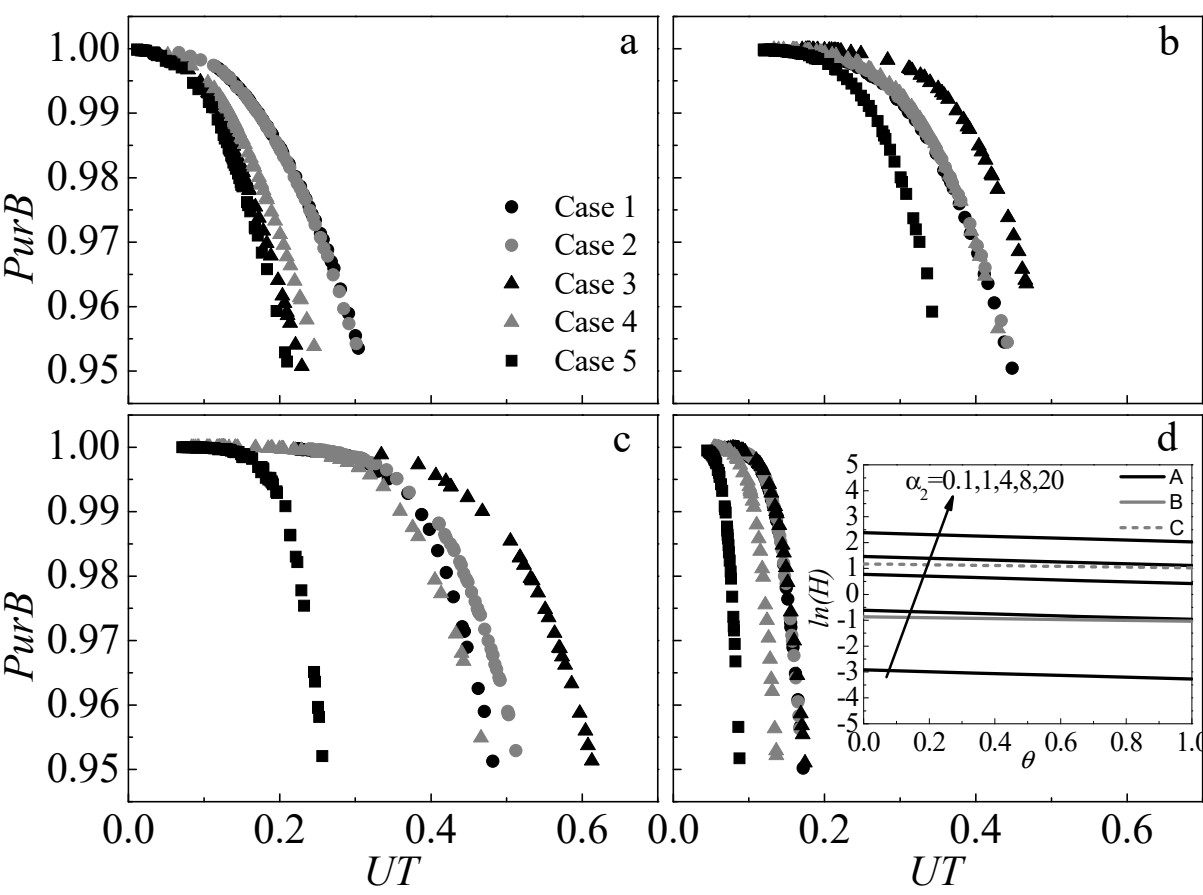

**Figure 7.** Effect of $H_A$ on Pareto solutions for different cases. (**a–d**) are for $\alpha_2$ equal to 0.1, 4, 8 and 20, respectively. The insert shows Henry's constant as a function of $\alpha_2$. Grey solid line for $B$, grey dash line for $C$, black solid lines for $A$ with different $\alpha_2$.

### 4.2.1. Overall Effects of $\alpha_2$

Pareto solutions for different cases and $\alpha_2$ values of 0.1, 4, 8 and 20 are shown in Figure 7a–d. The results for $\alpha_2 = 1$ were similar to those in Figure 2a. For clearer comparison, $UT$ values corresponding to $PurB = 0.99$ are summarized in Figure 8a. It is seen that, for all cases, the optimal $UT$ first increased and then decreased with increased $\alpha_2$ values. Cases 1–4 had their maxim at $\alpha_2$ around 8 whereas Case 5 had the maxim at $\alpha_2$ around 4. As shown in Figure 8b, conversion in Zone III ($conv_{III}$ defined in Equation (25)) decreased with increased $\alpha_2$ up to $\alpha_2 = 8$ and slightly increased when $\alpha_2$ was further increased to 20. Similar to the results discussed in the last section, the balance of conversion was mainly attributed to Zone II. Conversion in Zones I and IV is negligible. Figure 9 shows effects of $\alpha_2$ on $m$ values corresponding to the Pareto solutions for Cases 1. The results for other cases are shown in Supplementary Materials as Figure S4.

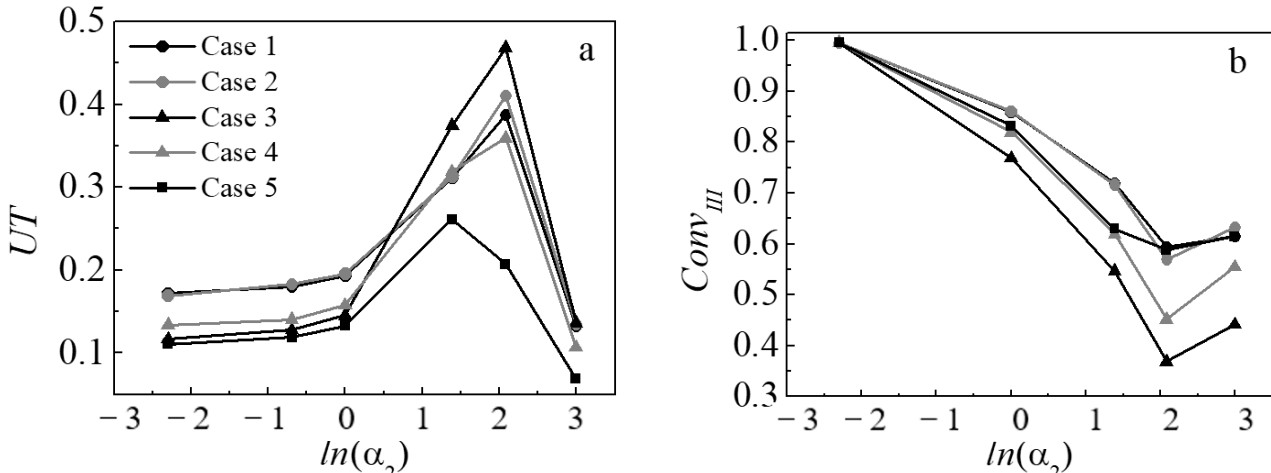

**Figure 8.** Maximal unit throughput (**a**) and conversions in zone III (**b**) as functions of $\alpha_2$. *PurB* = 0.99.

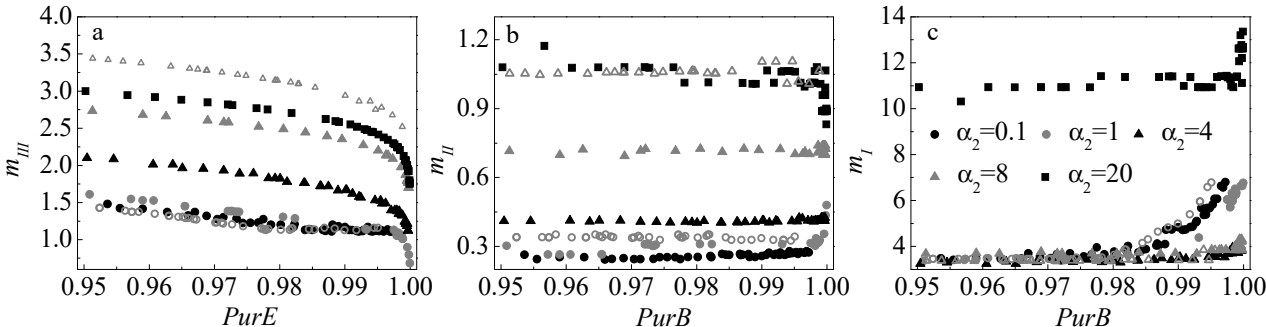

**Figure 9.** Effects of $\alpha_2$ on optimized *m* values. Solid points are for Case 1; hollow points are for Case 3 ($\alpha_2$ equal to 1 and 8). (**a**): effects of $\alpha_2$ on $m_{III}$; (**b**): effects of $\alpha_2$ on $m_{II}$; (**c**): effects of $\alpha_2$ on $m_I$.

### 4.2.2. $\alpha_2$ Lower than 1 ($H_A < H_B < H_C$)

It is seen in Figure 9a that, when $\alpha_2$ was decreased to 0.1, $m_{III}$ of Case 1 decreased in the low purity range (*PurB* < 0.98). This, according to Equation (28), increases the residence time in Zone III and is therefore favorable for converting more reactant (see Figure 8b). On the other hand, while the conversion in Zone II was decreased, $m_{II}$ was not increased but slightly decreased (Figure 9b). Lower $m_{II}$ was favorable to unit throughout. Since $m_{II}$ values were at the level of 0.3, much lower than the level of about 1.5 for $m_{III}$, the residence time in Zone II was anyway enough for conversion. Therefore, the reduction of flowrate in Zone II was mainly limited by its functional role in adsorptive separation, i.e., sufficiently carrying reactant *A* and product *B* to the feed port. Due to the decreased adsorption strength of *A* and less formation of *B*, the optimal $m_{II}$ values were slightly lower than those for $\alpha_2$ = 1. Figure 9c shows that, in the low purity range, $m_I$ was barely affected by the decrease of $\alpha_2$ from unity. Since the decrease in $m_{III}$ was more significant than the decrease in $m_{II}$, the overall unit throughput was decreased. In the high purity range beyond 0.98, increased $m_I$ further contributed to the reduced *UT*.

Figure 10 shows the internal concentration profiles under optimal conditions for Case 1 and purity of 0.99. It is seen in Figure 11a that, at this high purity, byproduct *C* was essentially purged out from Zone I by $0.7\tau_s$. It is due to the requirement of increased residence time in Zone III to further enhance the conversion for which the switching time and $m_I$ need to be increased (Equation (28)).

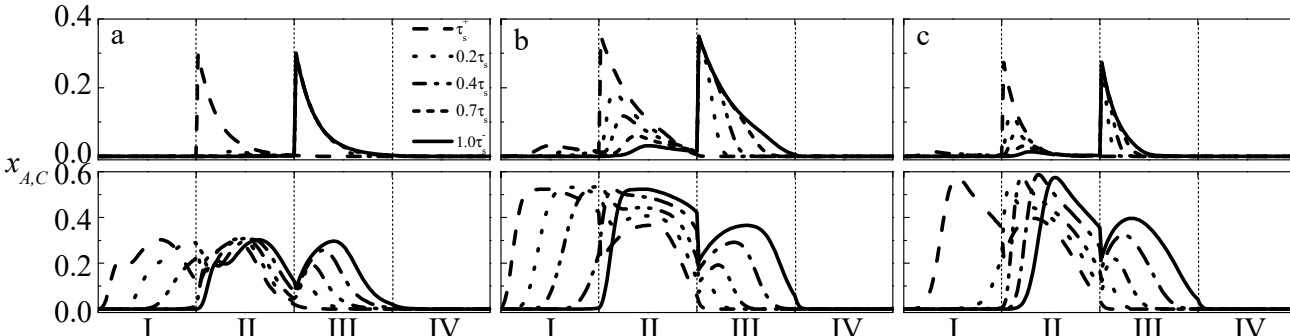

**Figure 10.** Internal concentration profiles at optimized conditions for various $\alpha_2$ values. Case 1, $PurB$ = 0.99. (**a**–**c**) for $\alpha_2$ equal to 0.1, 8 and 20, respectively. Upper for $A$ and lower for $C$.

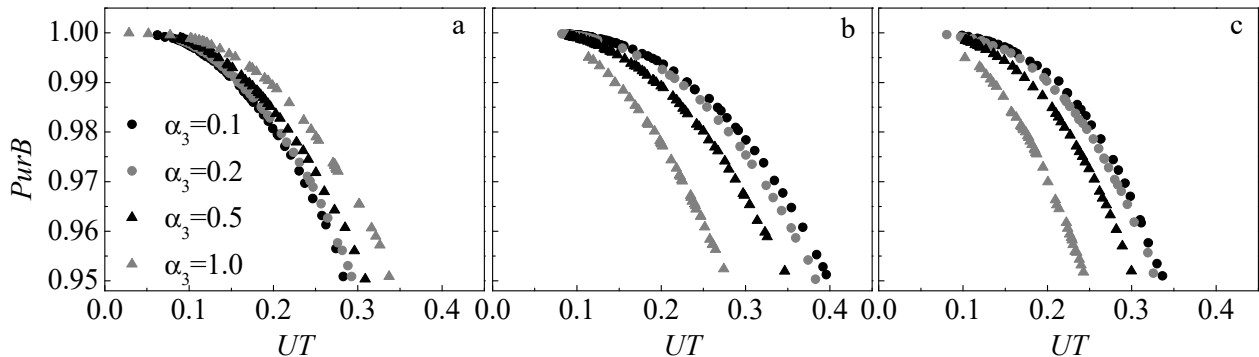

**Figure 11.** Effects of activation energy on Pareto curves for Cases 1 (**a**), 3 (**b**) and 5 (**c**).

As shown in Figure S4, $\alpha_2$ decreased to 0.1 had qualitatively similar effects on $m$ values for other cases, except that $m_I$ started to increase at different purity levels. Specifically, $m_I$ of Case 3 was increased in the whole purity range of interests. Pareto curves and corresponding $m$ values for $\alpha_2$ = 0.5 are shown in Supplementary Materials as Figure S5). For all cases, $UT$ decreased with $\alpha_2$ decreased from unity. However, as shown in Figures 7a and 8a, the effect was insignificant and did not change the sequence of comparison among various cases.

### 4.2.3. $\alpha_2$ in the Middle Range ($H_B < H_A \leq H_C$)

As $\alpha_2$ was increased to 4, corresponding to the condition of $H_B < H_A < H_C$, the Pareto curves of all cases were significantly shifted towards the right-up side (compare Figure 7b with Figure 2a). Due to the increased $H_A$, more fed reactant was retained in the solid phase in Zone III and needs to be further converted after being switched to Zone II. Conversion in Zone III was significantly reduced (from about 0.85 to about 0.65 for $PurB$ = 0.99). The reduced requirement of conversion allowed for an increase in $m_{III}$ (Figure 9a), favorable to the unit throughput. On the other hand, as the above mentioned in the last section, residence time in Zone II was anyway enough for the conversion. Increased $\alpha_2$ results in increased adsorption strength of $A$ and more formation of $B$ in Zone II. To efficiently convey A and B to the feed port, an increase in $m_{II}$ was required (Figure 9b). Since the conversion in Zone III was reduced, as $\alpha_2$ was increased to 4, an increase in $m_I$ was no longer required to increase the residence time. Therefore, $m_I$ is mainly determined by sufficient purge of byproduct $C$. As shown in Figure 10c, $m_I$ values were overlapping with those for lower $\alpha_2$ in the low purity range. This level of about 3.5 was barely affected by purity even in the high range, which was different from the trend of increased $m_I$ in high purity range for lower $\alpha_2$ values (see Fiugres 9c and S5). While both $m_{III}$ and $m_{II}$ increased with $\alpha_2$ increasing to 4, the former was more significant than the latter, resulting in improved unit

throughput. The improvement was further reinforced in the high purity range due to the relatively lower $m_I$ value compared with those of low $\alpha_2$ values.

As $\alpha_2$ was increased to 8, corresponding to the condition of $H_B < H_A \approx H_C$, Pareto curves and $m$ values exhibited qualitative trends similar to those of $\alpha_2 = 4$. Conversion in Zone III was further reduced and, accordingly, unit throughput was further improved for all cases except Case 5. The exceptional drop in *UT* for Case 5 will be explained in Section 4.2.4.

Different from the above discussed variation of $\alpha_2$ below unity, increased $\alpha_2$ in the middle range had significant effects on unit throughput. The significance was quantitatively different for different cases. As a result, orders of the optimal results upon comparison were changed. When $\alpha_2$ was increased to 4, Case 3 that had the largest temperature difference between Zones II and III gave the highest *UT*. When $\alpha_2$ was further increased to 8, the superiority of Case 3 was retained. Case 4, the other case with different $\theta_{II}^{\infty}$ and $\theta_{III}^{\infty}$, also became superior to Cases 1 and 2 (see Figure 7b,c).

For easy comparison, $m$ values optimized for Case 3 and $\alpha_2$ equal to 1 and 8 are also plotted in Figure 9 (hollow points). It is seen that, as $\alpha_2$ was increased to the middle range, $m_I$ and $m_{II}$ of Case 3 were similar to those of Case 1. The higher unit throughput of non-isothermal cases was therefore mainly attributed to the more significant increase in $m_{III}$. In addition to conversion of the reactant, SMBR has another important function of on-site adsorptive separation. More specifically, reactant *A* and byproduct *C*, both more preferentially adsorbed than product *B* as $\alpha_2$ was increased to the middle range, must be retained in Zone III to avoid polluting the product. It may be seen from Figure 10 that, for Case 1 and all $\alpha_2$ values, the SMBR was optimized at the operating conditions such that byproduct *C* was just about breakthrough from Zone III by the end of a switch. The similar behavior was also observed for other cases (Figure S6). Therefore, the increase in $m_{III}$, resulting from reduced conversion in Zone III, was limited by the requirement of effective adsorptive separation. Compared with Case 1, Case 3 had a lower temperature in Zone III, corresponding to higher adsorption strength, allowing for further increased $m_{III}$.

From the point of view of the overall process, both reaction and adsorptive separation were mainly realized in Zones II and III. A high temperature corresponded to fast kinetics and reduced adsorption strength. While a fast reaction rate was favorable in both zones for this kinetically controlled system, the reduced adsorption strength was desired only in Zone II but not in Zone III. Therefore, effects of temperature on reaction and separation were synergetic in Zone II but contradicting in Zone III. As $\alpha_2$ was increased to the middle range, more fed reactant was retained in the solid phase in Zone III and needs to be further converted in Zone II during the next switch. Due to the synergetic effects of temperature on kinetics and separation in Zone II, Cases 3 and 4 with a higher temperature in Zone II than in Zone III become superior to isothermal Cases 1.

### 4.2.4. $\alpha_2$ Equal to 20 ($H_B < H_C < H_A$)

As $\alpha_2$ is further increased to 20, adsorption strength reactant *A* is greater than those of both product *B* and byproduct *C*. Figure 9 shows that $m_{II}$ and $m_{III}$ kept increasing, following the previous trends. However, the optimal $m_I$ values are remarkably increased from about 3.5 to 13.5, which according to Equation (26), results in a significant drop of unit throughput. It is seen in Figure 10c that byproduct *C* is essentially purged out from Zone I before $0.7\tau_s$. In addition, in both Zones II and III, reactant *A* is continually propagating within the whole switch, different from the approaching of steady profile for $\alpha_2$ values lower than unity. At the end of a switch, the profile of A is developed only to about half of the columns in Zones II and III. Therefore, the increase in optimized $m_I$ was not for sufficient conversion but to purge unreacted *A*, which becomes the most preferentially adsorbed component at $\alpha_2 = 20$, out from Zone I. Increasing reactant adsorption strength beyond the value of heavy byproduct had a significant effect on optimal unit throughput. At $\alpha_2 = 20$, *UT* was remarkably reduced to the level even lower than that of $\alpha_2 = 1$.

Recall that $UT$ of Case 5 was decreased as $\alpha_2$ was increased from 4 to 8 (see Figures 7 and 8a), different from the other cases. The explanation to this exception was that Case 5 was the only case with $\theta_I^\infty = 0$. As shown in Figure 7, at this lowest temperature considered in the current work, $H_A$ became obviously greater than $H_C$ as $\alpha_2$ was increased to 8. Optimized $m_I$ was accordingly increased (see Figure S4 in Supplementary Materials), resulting in reduced $UT$, similar to the trends of other cases at $\alpha_2 = 20$.

Shown in Figure 8b was another notable feature at $\alpha_2 = 20$: conversion of $A$ in Zone III was increased. The explanation was that reactant $A$ was developed to only half of the column in Zone III during a switch. Although the adsorption strength of $A$ was increased, a reduced overall amount of $A$ was retained in Zone III and further converted in Zone II during the next switch. Since conversion in the other two zones was negligible, conversion in Zone III need to be increased accordingly.

### 4.3. Effects of Activation Energy

In this section, the effects of activation energy of forward reaction were investigated. A multiplier $\alpha_3$ with several values less than 1 was applied to decrease $e_f$ and the optimization problem was thereafter solved.

Figure 11 shows the Pareto curves for Cases 1, 3 and 5 and different $\alpha_3$ values. According to Equation (6), temperature dependence of the forward reaction rate was determined by the coupling of activation energy and a reference temperature, $\theta^{ref} = 0.667$. $Da$ was fixed at this reference temperature and became less sensitive to temperature with decreased $\alpha_3$. Case 1 was isothermally operated at $\theta_{I,II,III,IV}^\infty = 1$, higher than $\theta^{ref}$. Therefore, $Da$ decreased with deceased $\alpha_3$ and the unit throughput was accordingly decreased. On the other hand, when $\alpha_3$ was decreased, Case 5 that was operated at the lowest temperature had increased $Da$ and unit throughput. Case 3 had a high temperature in Zone II and a low temperature in Zone III. The $\alpha_3$ effects on Pareto solutions were qualitatively similar to those of Case 5 indicates kinetics play a more important role in Zone III than in Zone II, in agreement with previous discussions. An interesting observation is that, at reduced $\alpha_3$ values, Case 3 had unit throughput even higher than that of Case 1 at $\alpha_3 = 1$, by about 20%. A comparison of corresponding $m$ values (Supplementary Materials Figure S7) shows that this superiority was mainly attributed to the increased $m_{III}$. As aforementioned in the last section, temperature has contradicting effects on kinetics and adsorptive separation in Zone III. When the reaction rate became less sensitive to temperature with reduced $\alpha_3$, the low temperature in Zone III shows its beneficial effects on adsorption, resulting in higher $UT$. Effects of $\alpha_3$ on Pareto curves for Cases 2 and 4 were similar to those for Cases 1 and 3, respectively. The corresponding results were shown in Supplementary Materials as Figure S8.

### 4.4. Effects of Feed Concentration and Reaction Equilibrium

A multiplier of $\alpha_4$ was used to adjust $x_{feed}$ and to investigate its effects on the Pareto solutions. As shown in Figure S9, the Pareto solutions were shifted towards the left-hand side with increased feed concentration. This decrease in unit throughput is attributed to the increased reverse reaction rate, the only nonlinear term in the component balance equation. However, as aforementioned in previous discussions, under optimized conditions, the on-site separation of product $B$ and byproduct $C$ was efficient and the reverse reaction was much slower than that of the forward reaction. As a result, feed concentration increased by a factor up to 10 had minor effects on the dimensionless unit throughput. The order of various cases upon comparison was not changed.

Effects of reaction enthalpy were also investigated by applying a multiplier of $\alpha_5$. Both positive and negative values were assigned to $\alpha_5$, accounting for exothermic and endothermic reactions. Due to the efficient separation and negligible reversible reaction rate under optimized conditions, varied reaction enthalpy had minor effects on the obtained Pareto solutions. The results are provided in Supplementary Materials as Figure S10.

## 5. Conclusions and Remarks

The ongoing work is to evaluate and compare a total of five representative SMBR operation modes with different temperature distributions based on multiobjective optimization results. In this article, simultaneously maximization product purity and unit throughput of a reversible reaction $A \leftrightarrow B + C$ was considered. The effects of five model parameters, namely, reaction rate constant, Henry's constant of reactant, activation energy, feed concentration and reaction enthalpy, on the optimization results were systematically investigated by the application of corresponding multipliers. For more generality, a dimensionless model was derived and used in the non-isothermal SMBR simulations.

The results show that both reaction kinetics and adsorptive separation of products play important roles in the 4-zone SMBR operated under optimized conditions. Temperature effects on kinetics and adsorption were synergetic in Zone II but contradicting in Zone III. The original parameters obtained for methyl acetate synthesis had two important features: reaction rate was slow ($Da^{ref}$ = 2.5); adsorption strength of the reactant was close to that of the product but significantly lower than that of the byproduct. In this case, about 85% of the reactant was converted in Zone III and the system was kinetically controlled. High temperature in Zone III was therefore desired and Case 1, isothermally operated at the highest temperature, performed better than the other cases. With the increased reaction rate ($Da^{ref} \approx 10$), the non-isothermal cases with different temperatures in Zones II and III become superior to Case 1 due to the improved adsorptive separation. As $Da^{ref}$ was increased to about 200, the system became completely controlled by separation. Maximum uni throughput values approached those obtained for SMB separation of product and byproduct.

When adsorption strength of reactant was reduced to be lower than that of the product, conversion in Zone III was increased. Accordingly, unit throughput of the kinetically controlled system was reduced but the effects were secondary. When adsorption strength of reactant was increased to the middle range, such that $H_B < H_A \leq H_C$, unit throughput of all cases were significantly increased. Under this condition, a considerable fraction (40–65%, varied with cases) of reactant was converted in Zone II. Due to the synergetic effects of temperature on kinetics and adsorption in Zone II, non-isothermal operation cases became superior to isothermal cases. However, as $H_A$ was further increased and the reactant became the most preferentially adsorbed species, unit throughput of the SMBR was remarkably decreased by about 50% when compared with the results obtained for original parameters.

With reduced activation energy, the kinetics became less sensitive to temperature. As a result, non-isothermal operations favorable to adsorptive separation could be used to enhance the unit throughput. Under optimized conditions for the original kinetically controlled system, product and byproduct could be successfully separated in the SMBR. Rate of the reverse reaction was much lower than that of forward reaction. Therefore, feed concentration and equilibrium constant had minor effects on the optimization results.

Unit throughput is a function of $m_I$, $m_{II}$ and $m_{III}$. They had combined effects on the comparison among different operation modes and model parameters. The relative significance of these $m$ values depended not only on the operation mode and range of parameters but also on the range of product purity, the other objective defined in the optimization problem. The trends could not be directly predicted by the equilibrium theory, which has been extensively used for the design and analyses of SMB processes.

In all, when the reaction rate is fast ($Da^{ref}$ greater 10) or the adsorption strength of reactant is greater than that of product, nonisothermal operation may be applied to significantly enhance the unit throughput of an SMBR for reversible $A \leftrightarrow B + C$ reaction.

The results presented in this article were limited to the problem of simultaneously optimized unit throughput and product purity. It was acknowledged that, in addition to these two objectives, some other objective functions, such as solvent consumption, might be of great importance in many realistic process developments. Another practically important optimization problem, simultaneous maximization of purity and minimization

of solvent consumption at fixed unit throughput, was also investigated by this group. It was shown that reactant adsorption in the middle range was also favorable to the reduction of solvent consumption. More systematic and detailed results can be found in Wang's Master Thesis [50].

**Supplementary Materials:** The following are available online at https://www.mdpi.com/2227-9717/9/2/360/s1, Figure S1: Temperature transition in Zone II during a switch (Case 3), Figure S2: $m_{IV}$ values corresponding to simultaneously maximized PurB and UT (Cases 1, 3 and 5, $\alpha_1$ equal to 1 and 100), Figure S3: Internal concentration profiles of product B corresponding to PurB = 0.99, Figure S4: Effects of $\alpha_2$ on m values corresponding to the Pareto solutions for Cases 2, 3, 4, 5, Figure S5: Comparison of $\alpha_2$ equal to 0.5 and 1, Figure S6: Internal concentration profiles of byproduct C at optimal conditions for PurB = 0.99, $\alpha_2$ = 4, Cases 2, 3, 4, and 5, Figure S7: Comparison of m values optimized for Case 3/$\alpha_3$ = 0.5 and Case 1/$\alpha_3$ = 1, Figure S8: Effects of $\alpha_3$ on Pareto curves for Cases 2 and 4, Figure S9: Pareto curves obtained at various feed concentrations, Figure S10: Pareto curves obtained for various reaction enthalpy, Table S1: Dimensional model parameters, Table S2: Preset upper and lower bounds of variables for the use of NSGA.

**Author Contributions:** Conceptualization, J.X., W.Y. and A.R.K. Ray; methodology, J.X., W.Y. and A.R.K. Ray; software, J.W. and W.C.; validation, W.C. and Y.L.; investigation, J.X. and A.R.K. Ray; writing—original draft preparation, J.W. and J.X.; writing—review and editing, W.Y. and A.R.K. ray; visualization, J.W. and W.C.; supervision, J.X. and A.R.K. Ray; funding acquisition, J.X. and W.Y. All authors have read and agreed to the published version of the manuscript.

**Funding:** This research was funded by Zhejiang Provincial Natural Science Foundations, grant number LY19B060034, LY18B060008.

**Informed Consent Statement:** Informed consent was obtained from all subjects involved in the study.

**Data Availability Statement:** Not applicable.

**Conflicts of Interest:** The authors declare no conflict of interest.

## Nomenclatures

| | |
|---|---|
| *c* | mobile phase concentration, mol m$^{-3}$ |
| *Conv* | conversion |
| *d* | column diameter, m |
| $D_{app}$ | apparent dispersion coefficient in ED model, m$^2$ s$^{-2}$ |
| *Da* | Damkohler number |
| $E_f$ | activation energy, kJ mol$^{-1}$ |
| $e_f$ | dimensionless activation energy |
| *H* | Henry constant |
| $k_f$ | forward rate constant, s$^{-1}$ |
| $K_{eq}$ | equilibrium constant, mol m$^{-3}$ |
| $\widehat{K}_{eq}$ | dimensionless equilibrium constant |
| *L* | column length, m |
| *m* | flow rate ratio |
| *N* | plate number |
| *Pe* | Peclet number |
| *PurB* | product purity |
| *Q* | flow rate, m$^3$ s$^{-1}$ |
| $\widehat{Q}$ | dimensionless flow rate |
| *r* | reaction rate, mol m$^{-3}$ s$^{-1}$ |
| $\widehat{r}$ | dimensionless reaction rate |
| *R* | universal gas constant, 8.314 J mol$^{-1}$ K$^{-1}$ |

| | |
|---|---|
| $t$ | time, s |
| $T$ | temperature, K |
| $t_s$ | switching time, s |
| $UT$ | unit throughput |
| $V_{col}$ | column volume, m$^3$ |
| $x$ | dimensionless mobile phase concentration |
| $YB$ | yield |
| $z$ | axial coordinate, m |
| $Z$ | dimensionless axial coordinate |
| Greeks | |
| $\alpha_1$ | multiplier of forward rate constant |
| $\alpha_2$ | multiplier of Henry constant of reactant $A$ |
| $\alpha_3$ | multiplier of activation energy |
| $\alpha_4$ | multiplier of feed concentrations |
| $\alpha_5$ | multiplier of reaction enthalpy |
| $\Delta H_{ads}$ | adsorption enthalpy, kJ mol$^{-1}$ |
| $\Delta h_{ads}$ | dimensionless adsorption enthalpy |
| $\Delta H_{rxn}$ | reaction enthalpy, kJ mol$^{-1}$ |
| $\Delta h_{rxn}$ | dimensionless reaction enthalpy |
| $\varepsilon$ | column voidage |
| $\varphi$ | phase ratio |
| $\lambda$ | characteristic value for temperature transition, s$^{-1}$ |
| $\widehat{\lambda}$ | dimensionless value characterizing temperature transition |
| $\nu$ | stoichiometric number |
| $\theta$ | dimensionless temperature |
| $\theta^{rel}$ | relative temperature |
| $\tau$ | dimensionless time |
| $\tau_s$ | dimensionless switching time |
| Subscripts and superscripts | |
| *feed* | feed stream |
| $i$ | component index, $A$, $B$, $C$ for reactant, product and byproduct |
| $j$ | index of zones, $j = I, II, III, IV$ |
| $M$ | mash point along axial direction |
| *max* | highest value |
| *min* | lowest value |
| *ref* | reference temperature |

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
