# Peer review of "Multi-Objective Optimizations of Non-Isothermal Simulated Moving Bed Reactor: Parametric Analyses"

_processes, doi:10.3390/pr9020360_

Round 1
Reviewer 1 Report
The manuscript does not clearly state the interest of the research or its objectives.
The introduction does not provide relevant information.
The work methodology does not follow an obvious strategy.
In my opinion, the manuscript must be rewritten so that the reader of the journal Processes can understand the objective of the work, be able to understand the results obtained and compare with other works in this field.
Author Response
Reviewer 1:
The manuscript does not clearly state the interest of the research or its objectives.
The introduction does not provide relevant information.
The work methodology does not follow an obvious strategy.
In my opinion, the manuscript must be rewritten so that the reader of the journal Processes can understand the objective of the work, be able to understand the results obtained and compare with other works in this field.
Response:
To highlight the objectives and methodology of this work, the last paragraph of Introduction is re-written as follow:
In this article, the scope of the work is not limited to a specific chemical reaction but is extended to any reversible reactions in the form of A ↔ B+C. The objective of this study is to evaluate the feasibility of application of temperature gradient in SMBR system. More specifically, it is aimed at finding answers to the following two questions, which are of great academic and industrial interests: (a) What kind of kinetic and equilibrium properties should a reversible reaction of type A ↔ B + C must have such that non-isothermal operation mode can enhance the performance of a SMBR, and (b) How to adjust the operating parameters to meet the required objectives (productivity, purity, etc.) during the design of a non-isothermal SMBR process? To achieve these objectives, for the first time, a dimensionless mathematical model of SMBR for reversible reactions in the form of A ↔ B + C was developed for the simulation of non-isothermal SMBR processes. Subsequently, for more generality, effects of reaction rate, adsorption strength, activation energy, feed concentration and reaction enthalpy on the performance among various SMBR operation modes with different temperature distributions were systematically investigated based on multi-objective optimization results. To the best of our knowledge, this is the first attempt to apply dimensionless model and parametric analysis on the multi-objective optimization of non-isothermal SMBR processes. Hence, the results presented in this article is not restricted to methyl acetate synthesis but is valid for any reactions in the catalogue of A ↔ B + C. Furthermore, to provide deep insights to the trends of obtained optimization results and corresponding operating variables, additional informative results, including internal concentration profiles, conversion, and reaction rates in different zones, are discussed in detail.
Significant efforts have been made to improve the manuscript. The corrections and updates are highlighted in the revised manuscript.

Reviewer 2 Report
The paper gives an interesting analysis and modeling approach for multi-objective optimization of non-isothermal simulated Moving Bed Reactor.
I recommend to publish in present form.
Author Response
Reviewer 2
The paper gives an interesting analysis and modeling approach for multi-objective optimization of non-isothermal simulated Moving Bed Reactor.
I recommend to publish in present form.
Response: The authors like to thank the reviewer for finding the manuscript interesting with respect to modeling approach for multi-objective optimization of non-isothermal simulated moving bed reactor and its subsequent analysis.
Reviewer 3 Report
The structure of the submitted paper is a bit strange and I don't know what it wants to present. For example, in abstract section, there is no evidence that what has been exactly and which sodtware was used to simulate the SMBR, however, it needs more justification in the paper. Moreover, I'm not sure presenting the reaction (A>B+C) in the abstract is a good way to present it. It should be written more scientifically to be more understandable for readers.
My other comments are as follows;
1- In introduction section, you should compare the novelty of your work with previous literature and say what is the difference and why is important rather than works. I didn't see any comparison. The following references are recommended to cite and discuss as they focused on the reactors;
- A parametric study to simulate the non‐Newtonian turbulent flow in spiral tubes
- CFD design and simulation of ethylene dichloride (EDC) thermal cracking reactor
- Integrated feasibility experimental investigation of hydrodynamic, geometrical and, operational characterization of methanol conversion to formaldehyde
- Design and construction of a micro-photo bioreactor in order to dairy wastewater treatment by micro-algae: parametric study
- Thermodynamic Optimization of a Geothermal Power Plant with a Genetic Algorithm in Two Stages
2- Could you please provide numerical methods in more detail in section 2?
3- Please provide the number of nodes for numerical solution
4- Limitations of the work should be explained and discussed.
5- Does your simulation in 2D or 3D????
6- Section 3.3. should be explained in more detail as optimization is not adequate and I'm not sure what has done?
7- Pressure and temperature profile should be provided
8- Discussion should be improved.
Author Response
Reviewer 3
The structure of the submitted paper is a bit strange and I don't know what it wants to present. For example, in abstract section, there is no evidence that what has been exactly and which software was used to simulate the SMBR, however, it needs more justification in the paper. Moreover, I'm not sure presenting the reaction (A>B+C) in the abstract is a good way to present it. It should be written more scientifically to be more understandable for readers.
Response:
The following sentences are added in the abstract of the revised manuscript:
“Multi-objective optimization problems were solved by Non-dominated Sorting Genetic Algorithm. All calculations were carried out using FORTRAN codes.”
It is widely applied and accepted that chemical reactions can be generalized using representative letters for the indices of reactants and products. The expression of reaction is therefore kept emphasizing the article deals with all reactions of the type. However, the sentence in the abstract is corrected to avoid any misunderstanding.
Original: “Effects of five representative temperature distributions among different zones on the performance of a SMBR for reversiblereaction was evaluated based on simultaneously maximized unit throughput and product purity.”
Revised: “effects of five representative temperature distributions among different zones on the performance of an SMBR for reversible reaction in the general form of were evaluated based on simultaneous maximization of unit throughput and product purity.”
My other comments are as follows:
1- In introduction section, you should compare the novelty of your work with previous literature and say what is the difference and why is important rather than works. I didn't see any comparison. The following references are recommended to cite and discuss as they focused on the reactors.
- A parametric study to simulate the non‐Newtonian turbulent flow in spiral tubes
- CFD design and simulation of ethylene dichloride (EDC) thermal cracking reactor
- Integrated feasibility experimental investigation of hydrodynamic, geometrical and, operational characterization of methanol conversion to formaldehyde
- Design and construction of a micro-photo bioreactor in order to dairy wastewater treatment by micro-algae: parametric study
- Thermodynamic Optimization of a Geothermal Power Plant with a Genetic Algorithm in Two Stages
Response: Three of the above underlined papers are cited as Ref [40], [41] and [48] in the revised manuscript. The first two are related to parametric study and the last one is regarding optimization with genetic algorithm. While the other recommended papers are interesting, they are not directly related to the current study and, hence, not cited in this article.
The last paragraph of introduction section is re-written to highlight the novelty of this study.
“In this article, the scope of the work is not limited to a specific chemical reaction but is extended to any reversible reactions in the form of. The objective of this study is to evaluate the feasibility of application of temperature gradient in SMBR system. More specifically, it is aimed at finding answers to the following two questions, which are of great academic and industrial interests: (a) What kind of kinetic and equilibrium properties should a reversible reaction of typemust have such that non-isothermal operation mode can enhance the performance of a SMBR, and (b) How to adjust the operating parameters to meet the required objectives (productivity, purity, etc.) during the design of a non-isothermal SMBR process? To achieve these objectives, for the first time, a dimensionless mathematical model of SMBR for reversible reactions in the form of was developed for the simulation of non-isothermal SMBR processes. Subsequently, for more generality, effects of reaction rate, adsorption strength, activation energy, feed concentration and reaction enthalpy on the performance among various SMBR operation modes with different temperature distributions were systematically investigated based on multi-objective optimization results. To the best of our knowledge, this is the first attempt to apply dimensionless model and parametric analysis on the multi-objective optimization of non-isothermal SMBR processes. Hence, the results presented in this article is not restricted to methyl acetate synthesis but is valid for any reactions in the catalogue of. Furthermore, to provide deep insights to the trends of obtained optimization results and corresponding operating variables, additional informative results, including internal concentration profiles, conversion, and reaction rates in different zones, are discussed in detail”.
Significant efforts have been made to improve the manuscript. The corrections and updates are highlighted in the revised manuscript.
2- Could you please provide numerical methods in more detail in section 2?
Response: Numerical methods for SMB processes have been extensively discussed in the literature. The specific numerical scheme used in this group had been described in previous publications (Xu et al, AIChE J., 2013, 59, p4705-4714; Wang et al., Chem. Eng. J. 2020, 395, #125041). The following paragraph is added to Section 2.2, Numerical Solution. “In all, the original partial differential equation was first discretized by Martin-Synge method. The rigorous cyclic conditions were then replaced with initial conditions. As such, the model was converted to IVP and then solved using well established LSODA software. More details can be found elsewhere [31, 32].”
We believe that essential details of the numerical method have already been introduced in this manuscript.
3- Please provide the number of nodes for numerical solution
Response: The model used in this work is 1-dimenssional. As described in Section 2.2, “Martin Synge method divides a column into N equally-spaced sections and uses the 1st-order backward approximation for convection term (). If N is properly chosen, the truncation error can be used to eliminate the dispersion term ().” N was set to be 50 in this work. It was listed as a model parameter in Table 2.
4- Limitations of the work should be explained and discussed.
Response: As mentioned in Section 2.1 of the original manuscript, “It is acknowledged that the model had been developed based on several assumptions, especially the simple description of column temperature. These simplifications should not have qualitative effects on the conclusions in this study.”
The last paragraph in conclusion section is re-written as below:
“The results presented in this article are limited to the problem of simultaneously optimized unit throughput and product purity. It is acknowledged that, in addition to these two objectives, some other objective functions, such as solvent consumption, may be of great importance in many realistic process developments. Another practically important optimization problem, simultaneous maximization of purity and minimization of solvent consumption at fixed unit throughput, was also investigated by this group. It was shown that reactant adsorption in the middle range is also favorable to the reduction of solvent consumption. More systematic and detailed results can be found in Wang’s Master Thesis [51].”
- Does your simulation in 2D or 3D????
Response: The following descriptions are added in section 2.1 of the revised manuscript. “In our previous publications [31,32], conventional Equilibrium-dispersion (ED) model had been extended to account for the catalytic reaction. Due to the simplification of negligible radial gradients and instantaneous adsorption equilibrium, the reactor model is 1-dimensional. In addition, linear isotherm for all species was assumed.”
6- Section 3.3. should be explained in more detail as optimization is not adequate and I'm not sure what has done?
Response: The description of optimization problem is re-written as the following. “In general, high unit throughput, reduced solvent consumption, desired purity and recovery (yield) are the major objectives during the design of an SMBR process. One can consider many different configurations of SMBR for multi-objective optimization study. Several optimization problems with different combination of selections of objectives were investigated in our previous work [32]. This article is mainly focused on simultaneous maximization of product purity and unit throughput. In addition to these two objectives, constraints on purity and yield, greater than 0.95 and 0.90, respectively, were applied to practically narrow down the search space range in the operational range of parameters. A 4-zone SMBR has 5 independent operational parameters. In this work, flowrate in Zone I was fixed at the maximal flowrate, which is a scaling factor normally limited by column pressure [1, 4]. Following the notions of “Equilibrium theory” that has been extensively used in SMB design and analyses, the remaining 4 parameters were described by m-values (flow rate ratios) in the 4 zones. These m-values were all set to be the decision variables that can be independently tuned by an operator for simultaneous optimization of the selected objective fnctions. For clarity, the optimization problem is summarized in Table 4.”
Table 4. Optimization problem
Objectives |
Constraints |
Variables |
|
Decision |
Fixed |
||
Max PurB Max UT |
PurE > 0.95 YE > 0.90 |
mI, mII, mIII, mIV |
(Eqn 20) |
7- Pressure and temperature profile should be provided
Response: Pressure is not directly involved in the process modelling. However, maximum column pressure drop is limited (and related) to the highest flowrate allowed in Zone I. This maximum allowable flowrate is normally considered as a “Scaling Factor” for the design of liquid chromatography (and adsorption) processes. In the case of SMB, the maximum flowrate is normally assigned to Zone I (Nicoud, R.M., 2015, Chromatographic Processes: Modeling, Simulation and Design. Cambridge University Press; Jiang et al, J. 2018, Chromatogr. A, 1531, p131-142). In the dimensionless model, maximal flowrate (Qmax) was used to define several dimensionless variables (see Table 1).
Temperature was treated as in Eqn (9). As mentioned in Section 2.1 of the original manuscript, “It is acknowledged that the model had been developed based on several assumptions, especially the simple description of column temperature. These simplifications should not have qualitative effects on the specific conclusions in this study.”
8- Discussion should be improved.
Response: Several corrections have been made in the discussion section. They are highlighted in the revised manuscript. Some of the major corrections are summarized below.
In Section 4.1.2.
There are two quantitative differences in terms of m-values between the cases 1 and 3: (a) compared with Case 3, Case 1 has relatively lower mII and higher mIII, resulting in higher UT in the low purity range; and (b) mI of cases 1 and 3 starts to increase at PurB values of about 0.99 and 0.98 respectively. That mI of Case 1 starts to increase at a higher purity contributes to its high UT at high purity range.
“For all cases, the corresponding mIV values are scattered in a relatively large window with no obvious qualitative trends. This is in agreement with Eqn (26). Trends of optimized mIV are therefore omitted in the main manuscript but presented as Fig. S2 in Supplementary Materials for completeness.”
In Section 4.1.3:
“As shown in Figs. 4b and 4d, byproduct C is about to breakthrough for both cases by the end of a switch cycle. However, as the system is kinetically controlled and the reverse reaction is negligible compared with the forward reaction, the efficient separation of Case 3 has only minor effects on the optimal unit throughput. As a result of the negligible reverse reaction rate, steady profile of A can be approached despite the development of the other components. Given the established steady profile of A, further increase in mIII is limited by the sufficient retaining of by-product C that is more preferentially adsorbed than product B.”
In Section 4.2.2
“Lower mII is favorable to unit throughput. Since mII values are at the level of 0.3, much lower than the level of about 1.5 for mIII, the residence time in Zone II is anyway enough for conversion. Therefore, the reduction of flowrate in Zone II is mainly limited by its functional role in adsorptive separation, i.e., sufficiently carrying reactant A and product B to the feed port.”
“It is due to the requirement of increased residence time in Zone III to further enhance the conversion for which the switching time and mI need to be increased (Eqn. 28).”

Round 2
Reviewer 1 Report
The changes introduced by the authors have significantly improved the clarity of the objectives of the work and the discussion of the results. In my opinion, the manuscript can be published in present form.
Reviewer 3 Report
The paper can be published in the current version
This manuscript is a resubmission of an earlier submission. The following is a list of the peer review reports and author responses from that submission.
Round 1
Reviewer 1 Report
The authors developed a dimensionless model for a simulated moving bed reactor, based on their previous work, and evaluated the effects of selected parameters on the reactor dynamics. However, the model is the same as the reported one while the only difference is they used dimensionless parameters. Some of the parametric sensitivities were already conducted in their previous work (reference [32]), and the details of the multi-objective optimization study are similar. The second issue is the selection of the parameters. Some parameters selected are intrinsic, for example, the forward reaction rate constant, activation energy, and heat of reaction, which cannot be changed by the operating parameters. What is the purpose of the sensitivity analysis of those parameters? Other parameters include the plate number, which influences the numerical resolution, not the reactor dynamics. Therefore, due to the lack of novelty and motivation for the analysis, I am against the publication of the authors’ manuscript.
Reviewer 2 Report
The manuscript describes a numerical modelling study of a Simulated Moving Bed Reactor in which a model reaction A <-> B+C reaction takes place, along with adsosption/desorption, in a nonisothermal fashion. The study is conducted by setting up a multiobjective optimization problem, in a way that the authors call "non-restrictive optimization", operated over the four indicated design variables, essentially related to the flow rates in the four column units. My understanding of "non-restrictive" is that performance criteria are not pushed to the limit but relaxed in the attempt of attaining a more global optimum. The influence of various parameters is presented and discussed.
The system, although regarding a generic chemical process, is complex, as complex are the arguments in the discussion, and this reflects on the readability of the manuscript - however I must admit that I am not familiar with this type of process.
In detail:
0) The Abstract adequately summarizes the contents of the paper.
1) The Introduction is fine and adequately presents relevant literature, to my best knowledge.
2) The Mathematical model is clearly presented; so is the Numerical method, apart from a statement in the opening of the relevant subsection:
"The dispersion term (∂/∂z2) is replaced by the truncation error introduced by the 1st order backward approximation of the convection term (∂/∂z)",
followed by Eq.s (12) and (13). This statement is bizarre and must be rephrased or better explained. It appears that the dispersion term is not approximated but somehow left to the truncation error to represent it. But in this way there is no control over the magnitude of the numerical diffusion term thus introduced, that I think can only show as a qualitative relaxation of the spatial gradients.
3) The Optimization problem is rather clearly set up as a search of Pareto sets, after having reasonably reduced the feasibility domain by manipulating the constraints on purity and yield, in the attempt of maximizing product purity and unit throughput over four design variables. The method employed is based on the Non-dominated Sorting Genetic Algorithm (NSGA II). A final choice among the Non-dominated solutions found is not attempted, rather a parametric study is conducted as explained in Section 4.
4) The Results and Discussion section illustrates the optimization results as some parameters change, namely: the forward reaction rate, the adsorption strength of reactant A, the activation energy, the feed concentration and the reaction equilibrium.
The manuscript definitely deserves publication, after one major clarification, indicated above at point 2 (physical vs numerical dispersion), and minor language adjustments, some of which I suggest in the attached marked pdf file. I trust the authors should be able to operate themselves, having a native english language speaker in the team.

Reviewer 3 Report
The authors do not explain the reaction they want to model, or the type of reactor and its dimensions on which it is intended to be carried out. Perhaps a drawing of the reactor would be explanatory. It is very difficult for the reader to visualize the influence of the process variables on the result. This makes the discussion of the results unclear.
The manuscript does not clearly state the interest of the research or its objectives. The introduction does not provide relevant information. The work methodology does not follow an obvious strategy.